

# Identification of potential molecular mimicry in pathogen-host interactions

Kaylee D. Rich[1,2], Shruti Srivastava[1,2], Viraj R. Muthye[1,2] and James D. Wasmuth[1,2]

[1] Faculty of Veterinary Medicine, University of Calgary, Calgary, Alberta, Canada
[2] Host-Parasite Interactions Research Training Network, University of Calgary, Calgary, Alberta, Canada

## ABSTRACT

Pathogens have evolved sophisticated strategies to manipulate host signaling pathways, including the phenomenon of molecular mimicry, where pathogen-derived biomolecules imitate host biomolecules. In this study, we resurrected, updated, and optimized a sequence-based bioinformatics pipeline to identify potential molecular mimicry candidates between humans and 32 pathogenic species whose proteomes' 3D structure predictions were available at the start of this study. We observed considerable variation in the number of mimicry candidates across pathogenic species, with pathogenic bacteria exhibiting fewer candidates compared to fungi and protozoans. Further analysis revealed that the candidate mimicry regions were enriched in solvent-accessible regions, highlighting their potential functional relevance. We identified a total of 1,878 mimicked regions in 1,439 human proteins, and clustering analysis indicated diverse target proteins across pathogen species. The human proteins containing mimicked regions revealed significant associations between these proteins and various biological processes, with an emphasis on host extracellular matrix organization and cytoskeletal processes. However, immune-related proteins were underrepresented as targets of mimicry. Our findings provide insights into the broad range of host-pathogen interactions mediated by molecular mimicry and highlight potential targets for further investigation. This comprehensive analysis contributes to our understanding of the complex mechanisms employed by pathogens to subvert host defenses and we provide a resource to assist researchers in the development of novel therapeutic strategies.

## INTRODUCTION

The ability to disrupt host biochemical pathways is a crucial survival strategy employed by pathogenic species from across all kingdoms of life. These pathogens target a wide range of molecules involved in important processes, such as the host's cell cytoskeleton, signaling and immune system. Biomolecules, including proteins, lipids, and carbohydrates, are secreted by the pathogen, or displayed on its cell surface and play a role in mediating interference. These parasite-derived biomolecules may imitate host biomolecules, either in sequence, tertiary structure or both. This phenomenon is termed molecular mimicry and

Corresponding author
James D. Wasmuth,
jwasmuth@ucalgary.ca

was originally defined as the sharing of antigenic determinants between a pathogen and its host (*Damian, 1964*). There are two commonly observed consequences of molecular mimicry. One is manipulation of the host by the pathogen. The other is when a pathogen induces autoimmunity due to its antigens cross-reacting with host self-antigens (reviewed in *Rojas et al., 2018*). These two consequences should be considered to exist on a spectrum, rather than independently. For this study, we focus on the first, which directly benefits the pathogen and has been demonstrated across a broad range of pathogenic species.
For example, the parasitic protozoan *Toxoplasma gondii* secretes the dense granule protein GRA24, which is a potent modulator of the immune response. GRA24 contains a p38a docking motif that resembles those of the PTP tyrosine phosphatase family (*Braun et al., 2013*; *Mercer et al., 2020*). It is exported to the host cell nucleus and forms a complex with the host enzyme p38a, triggering the enzyme's autophosphorylation. Through this, GRA24 regulates the expression of a suite of cytokines, and triggers a strong Th1 response, which increases parasite survival by striking a balance in the immune response, promoting host survival (*Mercer et al., 2020*).

In addition to immune modulation, molecular mimicry is responsible for other host-pathogen interactions, including those involving human extracellular matrix and cell adhesion proteins. For example, the pathogenic fungi, *Candida albicans*, produces the agglutinin-like sequence protein 3 (Als3), which resembles the immunoglobulin domain of host E-cadherin (*Liu & Filler, 2011*). Through mimicry, Als3 promotes adhesion of host cells, biofilm formation, and endocytosis of *C. albicans* into the host cell. The pathogenic bacterium, *Helicobacter pylori* uses a type IV secretion system (T4SS) to inject virulence proteins into its host. A critical component of the *H. pylori* T4SS is a cytotoxin-associated gene protein (CagL), which interacts with host cell integrin, triggering subsequent delivery of virulence proteins into the host cell. CagL contains a RGD motif, mimicking host fibronectin, a natural ligand of integrin (*Kwok et al., 2007*).

For these and many examples, the resemblance between the pathogen and host proteins was typically discovered once a candidate mediator was identified. This bespoke process typically relied on sequence similarity alignments, *e.g.*, BLAST (*Altschul et al., 1990*). With genome assemblies increasingly commonplace, particularly for pathogenic species, it became feasible to compare entire proteomes to find potential mimicry. Arguably the first attempt was by *Ludin, Nilsson & Mäser (2011)*, who collected proteomes from eight species of pathogenic eukaryotes, human, and a further seven species of non-pathogenic eukaryotes. Through this pipeline they searched for unexpected similarity between the pathogen and human proteins. Briefly, the pathogens' proteins were searched against non-pathogens' proteins, which acted as a negative control, and those with no match were split into *k*-mers of length 14 amino acids (hereafter called 14-mers). The pathogens' 14-mers were again searched against the human and the non-pathogens' proteins. Those that matched better to human proteins were considered candidate regions of molecular mimicry. This approach has been modified and used across dozens of bacteria pathogens and for single pathogen-host comparisons (*Doxey & McConkey, 2013*; *Hebert et al., 2015*; *Armijos-Jaramillo et al., 2021*).

Unfortunately, the source code for the original *Ludin, Nilsson & Mäser (2011)* pipeline and the accompanying mimicDB are no longer available. In this current study, we had two goals. For the first, we recreated the pipeline to the best of our ability and validated it with *Plasmodium falciparum* (PLAF7). We then extended pipeline to include additional filtering steps. Our second goal was to search for mimicry candidates across 32 pathogens —bacteria, fungi, protozoa, and helminths—of global health importance. We selected these species because their proteomes were among the first to run through AlphaFold2 and have tertiary protein structures predicted (*Jumper et al., 2021*). We used this structural information to further filter mimicry candidates and identified common cellular pathways or functions potentially exploited through mimicry by different pathogens. We found relatively few potential mimics in most species of pathogenic bacteria, and high numbers in the pathogenic fungi and protozoans. In the host species, immune-related proteins were underrepresented as targets of mimicry, whereas proteins involved in host extracellular matrix organization and cytoskeletal processes were overrepresented. The number of mimicking regions identified was extremely sensitive to types of low complexity regions (LCRs), such as repeats of short oligopeptides.

## METHODS

The bioinformatic pipeline is our implementation of the one described by *Ludin, Nilsson & Mäser (2011)* (Fig. 1). We have extended the pipeline to include solvent accessibility and low complexity measurement. All acronyms and methods summarized in Fig. 1 are described in more detail below. All Python and Bash scripts used are available at https://github.com/Kayleerich/molecularmimicry and https://doi.org/10.5281/zenodo.8361282. Portions of this text were previously published as part of a preprint (https://doi.org/10.1101/2023.06.14.544818).

### Pipeline recreation validation

We compared our recreation of the sequence similarity and *k*-mer filtering steps (as described below) with the original pipeline to the best of our ability using *Plasmodium falciparum* downloaded from PlasmoDB (available here: https://plasmodb.org/common/downloads/release-56/Pfalciparum3D7/fasta/data/) and the control species specified by Ludin et al. (*Arabidopsis thaliana*, *Caenorhabditis elegans*, *Schizosaccharomyces pombe*, *Ciona intestinalis*, and *Trichoplax adhaerens*) (Table 1). Negative control species' and host (*Homo sapiens*) proteomes were downloaded from Uniprot reference proteomes (Release 2022_01) (*The UniProt Consortium, 2023*).

### Species selection

The selection of the 32 pathogen and 13 control species for this study was guided by data availability (Table 1). These 45 species, plus the host human species, were the first to have their proteomes run through AlphaFold2 software (*Jumper et al., 2021*; *Varadi et al., 2022*). The dataset for global health pathogens is available here: https://alphafold.ebi.ac.uk/download#global-health-section (2022). The dataset for the host and control species, aka model organisms, is available here: https://alphafold.ebi.ac.uk/download#proteomes-section (2022). The protein sequences for all species in this study were downloaded from

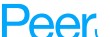

**Figure 1 The mimicry identification pipeline used in our analysis.** Descriptions for each step are included in the blue boxes.

**Table 1** The proteomes of species used in this study.

| Species name | Species description and relevance | Category | Uniprot ID | #Ref. protein |
|---|---|---|---|---|
| *Homo sapiens* | Human | Host* | UP000005640 | 20,577 |
| *Ajellomyces capsulatus* | Opportunistic pathogenic yeast, pulmonary histoplasmosis | Pathogen | UP000001631 | 9,214 |
| *Brugia malayi* | Filarial nematode, elephantiasis | Pathogen | UP000006672 | 8,825 |
| *Campylobacter jejuni* | Gram-negative bacterium, campylobacteriosis | Pathogen | UP000000799 | 1,623 |
| *Cladophialophora carrionii* | Melanized fungus, subcutaneous chromoblastomycosis | Pathogen | UP000094526 | 11,173 |
| *Dracunculus medinensis* | Guinea worm, dracunculiasis | Pathogen | UP000274756 | 10,868 |
| *Enterococcus faecium* | Gram-positive bacterium, opportunistic pathogen with multi-drug antibiotic resistance | Pathogen | UP000325664 | 3,119 |
| *Fonsecaea pedrosoi* | Fungus, subcutaneous chromoblastomycosis | Pathogen | UP000053029 | 12,525 |
| *Haemophilus influenzae* | Gram-negative bacterium, range of infections including meningitis and pneumonia | Pathogen | UP000000579 | 1,704 |
| *Helicobacter pylori* | Gram-negative bacterium, peptic ulcers | Pathogen | UP000000429 | 1,554 |
| *Klebsiella pneumoniae* | Gram-negative bacterium, range of healthcare-associated infections | Pathogen | UP000007841 | 5,728 |
| *Leishmania infantum* | Protozoan, visceral leishmaniasis | Pathogen | UP000008153 | 8,045 |
| *Madurella mycetomatis* | Fungus, mycetoma | Pathogen | UP000078237 | 9,733 |
| *Mycobacterium leprae* | Gram-positive bacterium, leprosy | Pathogen | UP000000806 | 1,603 |
| *Mycobacterium tuberculosis* | Gram-positive bacterium, tuberculosis | Pathogen | UP000001584 | 3,993 |
| *Mycobacterium ulcerans* | Gram-positive bacterium, buruli ulcers | Pathogen | UP000020681 | 9,033 |
| *Neisseria gonorrhoeae* | Gram-negative bacterium, gonorrhea | Pathogen | UP000000535 | 2,106 |
| *Nocardia brasiliensis* | Gram-positive bacterium, nocardiosis | Pathogen | UP000006304 | 8,414 |
| *Onchocerca volvulus* | Filarial nematode, river blindness | Pathogen | UP000024404 | 12,119 |
| *Paracoccidioides lutzii* | Fungus, paracoccidioidomycosis | Pathogen | UP000002059 | 8,811 |
| *Plasmodium falciparum* | Protozoan, malaria | Pathogen | UP000001450 | 5,376 |
| *Pseudomonas aeruginosa* | Gram-negative bacterium, opportunistic pathogen with multi-drug antibiotic resistance | Pathogen | UP000002438 | 5,564 |
| *Salmonella typhimurium* | Gram-negative bacterium, range of infections including gastroenteritis | Pathogen | UP000001014 | 4,533 |
| *Schistosoma mansoni* | Blood fluke, intestinal schistosomiasis | Pathogen | UP000008854 | 14,097 |
| *Shigella dysenteriae* | Gram-negative bacterium, shigellosis | Pathogen | UP000002716 | 3,897 |
| *Sporothrix schenckii* | Fungus, sporotrichosis | Pathogen | UP000018087 | 8,673 |
| *Staphylococcus aureus* | Gram-positive bacterium, opportunistic pathogen with methicillin resistance | Pathogen | UP000008816 | 2,889 |
| *Streptococcus pneumoniae* | Gram-positive bacterium, range of infections including pneumonia | Pathogen | UP000000586 | 2,030 |
| *Strongyloides stercoralis* | Threadworm, strongyloidiasis | Pathogen | UP000035681 | 12,823 |
| *Trichuris trichiura* | Whipworm, trichuriasis | Pathogen | UP000030665 | 9,625 |
| *Trypanosoma brucei* | Protozoan, sleeping sickness | Pathogen | UP000008524 | 8,561 |
| *Trypanosoma cruzi* | Protozoan, Chagas disease | Pathogen | UP000002296 | 19,242 |
| *Wuchereria bancrofti* | Filarial nematode, lymphatic filariasis | Pathogen | UP000270924 | 13,000 |
| *Arabidopsis thaliana* | Thale cress, model organism | Control* | UP000006548 | 27,473 |
| *Caenorhabditis elegans* | Nematode, model organism | Control* | UP000001940 | 19,818 |
| *Candida albicans* | Commensal yeast, model organism | Control | UP000000559 | 6,035 |
| *Danio rerio* | Zebrafish, model organism | Control | UP000000437 | 25,707 |
| *Dictyostelium discoideum* | Slime mold amoeba, model organism | Control | UP000002195 | 12,727 |

| Species name | Species description and relevance | Category | Uniprot ID | #Ref. protein |
|---|---|---|---|---|
| *Drosophila melanogaster* | Common fruit fly, model organism | Control | UP000000803 | 13,821 |
| *Escherichia coli* | Gram-negative bacterium, model organism | Control | UP000000625 | 4,402 |
| *Glycine max* | Soybean, model organism | Control | UP000008827 | 55,855 |
| *Methanocaldococcus jannaschii* | Thermophilic methanogenic archaean, model organism | Control | UP000000805 | 1,787 |
| *Oryza sativa* | Rice, model organism | Control | UP000059680 | 43,672 |
| *Saccharomyces cerevisiae* | Brewer's yeast, model organism | Control | UP000002311 | 6,059 |
| *Schizosaccharomyces pombe* | Fission yeast, model organism | Control* | UP000002485 | 5,122 |
| *Zea mays* | Corn, model organism | Control | UP000007305 | 56,926 |
| *Ciona intestinalis* | Sea squirt, model organism | Control** | UP000008144 | 16,680 |
| *Trichoplax adhaerens* | Placozoan, model organism | Control** | UP000009022 | 11,518 |

**Notes:**
* Species used for pipeline verification and molecular mimicry survey.
** Control species used for pipeline verification only.

Uniprot reference proteomes (Release 2022_01) (*The UniProt Consortium, 2023*). We discuss considerations for species selection later.

## Sequence similarity survey

Full-length pathogen proteins were queried against a database of control proteins using BLASTP (v2.12.0) (*Altschul et al., 1990*) and pathogen proteins with significant alignments (E-value < $1^{-10}$) to control proteins were removed. Phobius (v1.01) (*Käll, Krogh & Sonnhammer, 2004*) and bedtools (v2.30.0) (*Quinlan & Hall, 2010*) were used to identify and mask signal peptides in pathogen proteins. Pathogen proteomes were converted into short, overlapping sequences (*k*-mers) of 14 amino acids in length.

## *k*-mer filtering

Pathogen 14-mers were queried against control proteomes with ungapped BLASTP. Queries with high identity to a control sequence were removed (as described by *Ludin, Nilsson & Mäser, 2011*). Ungapped BLASTP was again used to compare remaining pathogen sequences to the host proteome. Sequences with high identity to a host sequence were retained.

## Pairing and merging alignments

Overlapping *k*-mers were merged into contiguous regions using a custom Python script (see Data availability section). As shown in Fig. 2, regions on the pathogen protein are referred to as mimicking pathogen regions (MPRs) and each corresponds to one or more regions on a host protein. Regions on a host protein are referred to as mimicked host regions (MHRs).

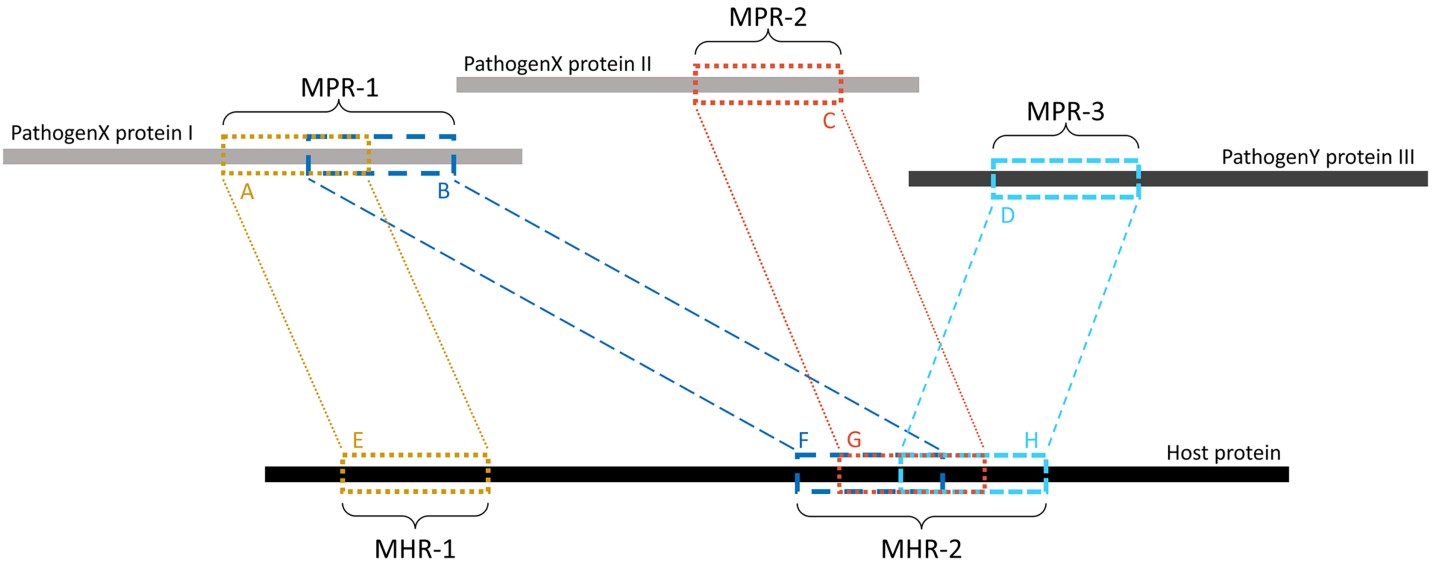

**Figure 2  Example of MPRs and MHRs, and how they relate to each other.** Each region from a pathogen protein is paired with a region from a host protein: region A is paired with region E. A mimicking pathogen region (MPR) is the region comprised of all overlapping 14-mers from a single pathogen protein that aligned to any host protein during pathogen-host sequence filtering: MPR-1 is comprised of regions A and B. A mimicked host region (MHR) is all merged corresponding regions on a host protein and multiple regions may overlap on the same protein: MHR-2 is made up of regions F, G, and H. Mimicry regions can be shared between one or more proteins: MPR-1 is shared between two MHRs on one host protein. MHRs can also be shared between pathogen species: MHR-2 is shared between three pathogen proteins and two pathogen species.

## Solvent-accessible surface area survey

The quotient of surface-accessible to total surface area (QSASA) for each amino acid residue in all pathogen-host paired regions was calculated using POPscomp (v3.2.1), where a value greater than 0.50 indicates that 50% of the residue is likely exposed on the surface of the protein (*Fraternali, 2002*). We retained host-pathogen sequence pairs if the mean QSASA for both pathogen and host regions was greater than 0.75.

## Protein clustering and gene ontology analysis

All human proteins containing an MHR pairing to at least one pathogen species protein were clustered using CD-hit (v4.8.1) (*Li & Godzik, 2006*; *Fu et al., 2012*). Gene Ontology enrichment analysis was done on proteins clustered at 100% in comparison to the full human proteome clustered at 100% using PANTHER Fisher's Exact Overrepresentation Test with false discovery rate correction (v17.0) (*Mi et al., 2019*; *Thomas et al., 2022*). Pathogen species associated with each term were identified as depicted in Fig. 3.

## Disorder prediction and low-complexity region filtering

Prediction of intrinsically disordered regions was done using IUPred3 (prediction type: short, smoothing used: medium) (*Erdős, Pajkos & Dosztányi, 2021*). Low complexity regions (LCRs) from all human proteins were identified using Segmasker (v 1.0.0, default parameters) (*Wootton & Federhen, 1996*; *Camacho et al., 2009*). MHRs that overlapped an LCR by more than 50% were removed using bedtools.
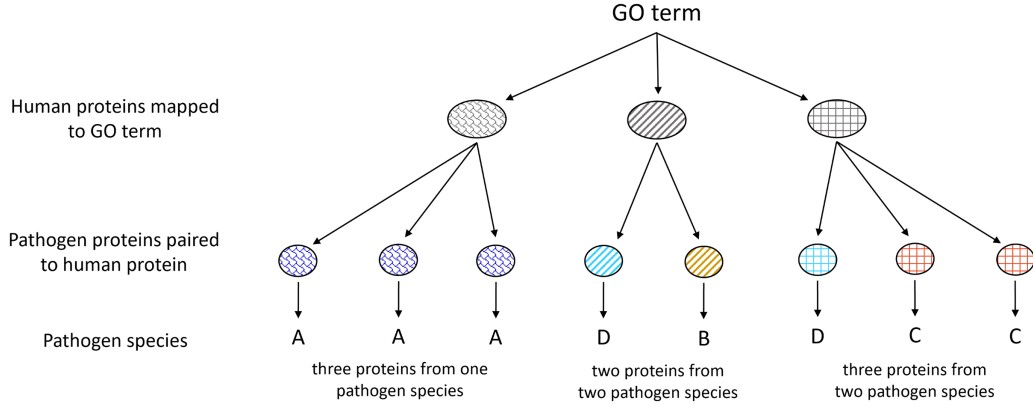

**Figure 3** **Illustration of Gene Ontology assignment between host and pathogen proteins.** Each host protein assigned to a given GO term may contain an MHR which aligns to MPRs in proteins from one or more pathogen species. The number of associated pathogen species is obtained through identifying the pathogen proteins paired to each host protein mapped to the given term.

## RESULTS

In this study, we implemented a sequence-based bioinformatics pipeline, as previously described by *Ludin, Nilsson & Mäser (2011)*, to identify potential molecular mimicry between humans and 32 pathogenic species of importance in global health. In brief, we removed proteins that were likely homologous between pathogens and non-pathogenic model species (aka negative controls), and then identified short regions that exhibited higher similarity between pathogen(s) and humans than the negative controls.

We compared the prediction of mimics in *P. falciparum* for the original pipeline and our interpretation. While we identified 85% more candidate 14mers in *P. falciparum* (544 to 1,014), these came from a comparable number of *P. falciparum* proteins (196 to 248; 25% more). It is important to note that the comparison is not just in the pipeline implementation, as there were changes in software versions (*e.g.*, BLAST) and proteome release versions. Further, certain algorithm parameters (*e.g.*, word-size, amino acid substitution matrix) were not in the original pipeline description.

### Shared sequences between pathogens and host

Across the 32 pathogenic species, we observed considerable variation in the number of potential mimics: between two and 4,093 14-mers from between one and 850 parasite proteins (Fig. 4). Notably, we found that most pathogenic bacteria had fewer 14-mer mimic candidates, even controlling for their smaller proteomes; the exceptions were *Mycobacterium tuberculosis* and *Mycobacterium ulcerans*. In comparison, helminths typically had fewer 14-mer mimic candidates compared to pathogenic protozoans and fungi. To identify unique protein regions that could potentially act as mimics, we merged overlapping 14-mer sequences into mimicking pathogen regions (MPRs) and mimicked

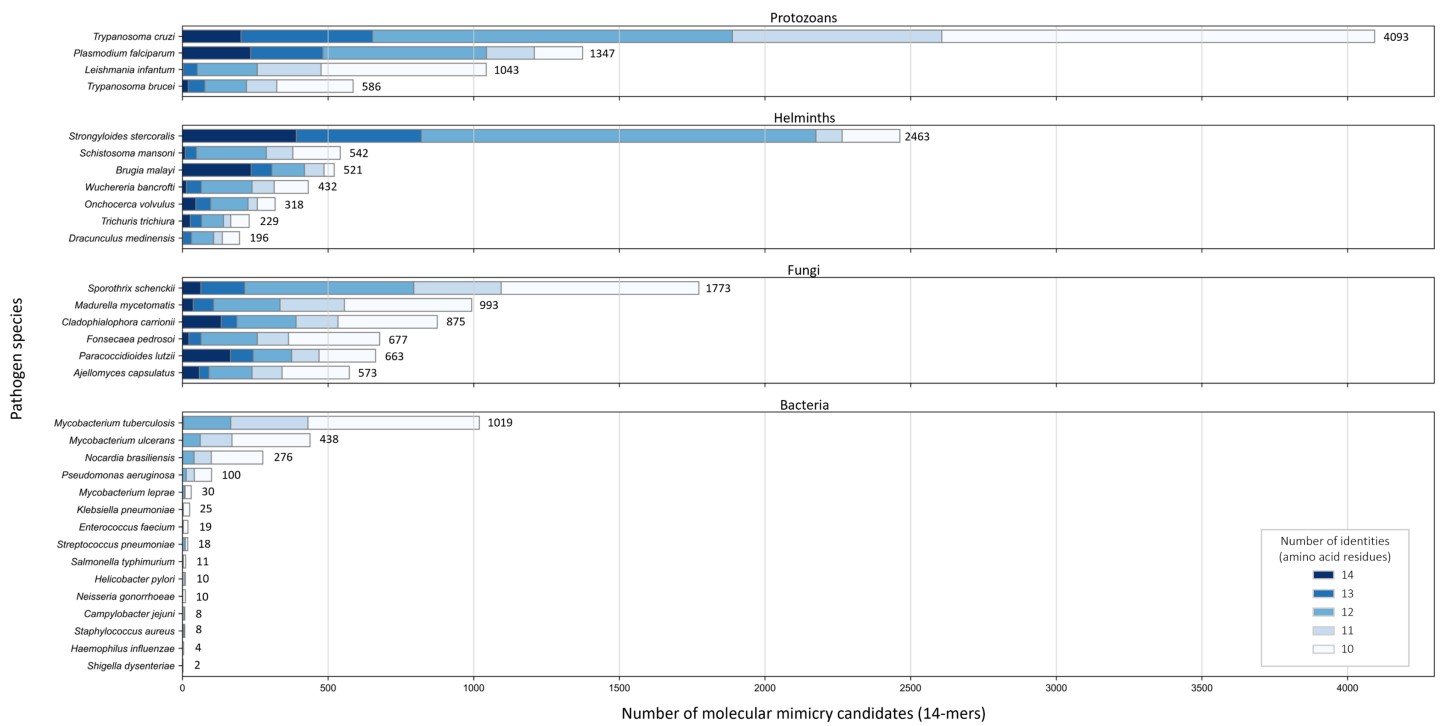

**Figure 4 Number of 14-mers with a hit to a human protein for each species coloured by group and number of identical amino acid residues.**

host regions (MHRs). Across the 32 pathogenic species, we identified a range of one to 1,368 MPRs and one to 1,006 MHRs (Fig. 5, blue).

Considering that host-pathogen interactions are likely mediated by motifs located on accessible surfaces rather than buried within the tertiary structure, we wanted to assess the solvent accessibility of the MHRs and MPRs to prioritize these regions for subsequent analysis. We compared the solvent accessibility of the candidate mimicry regions with randomly selected sequences from two baseline sets: (i) the complete proteome of each pathogen or host species, and (ii) peptides that had a high scoring pair (HSP) between pathogen and host protein. We found that that the candidate mimicry regions were enriched for solvent-accessible regions compared to both baseline sets for the host and 31 out of the 32 pathogens, with *M. tuberculosis* being the exception (Figs. 6, S1 and S2). For further analysis, we selected MHRs and MPRs with a mean QSASA > 0.75, which excluded all sequences for four bacterial species (Fig. 5, red).

## MHR-containing proteins

After applying QSASA filtering, we identified 1,878 MHRs in 1,439 human proteins. To investigate whether specific protein families or biological processes, such as immune response, were targeted by multiple pathogens, we clustered the human proteins at seven different stringencies, ranging from 100% identity to 40% identity. If there was an effect of gene family, we would expect to see an increase in the proportion of clusters that have MHRs to multiple pathogen species as the identity threshold of the clustering decreased. When human proteins were clustered at 100% identity, resulting in 1,438 clusters for the

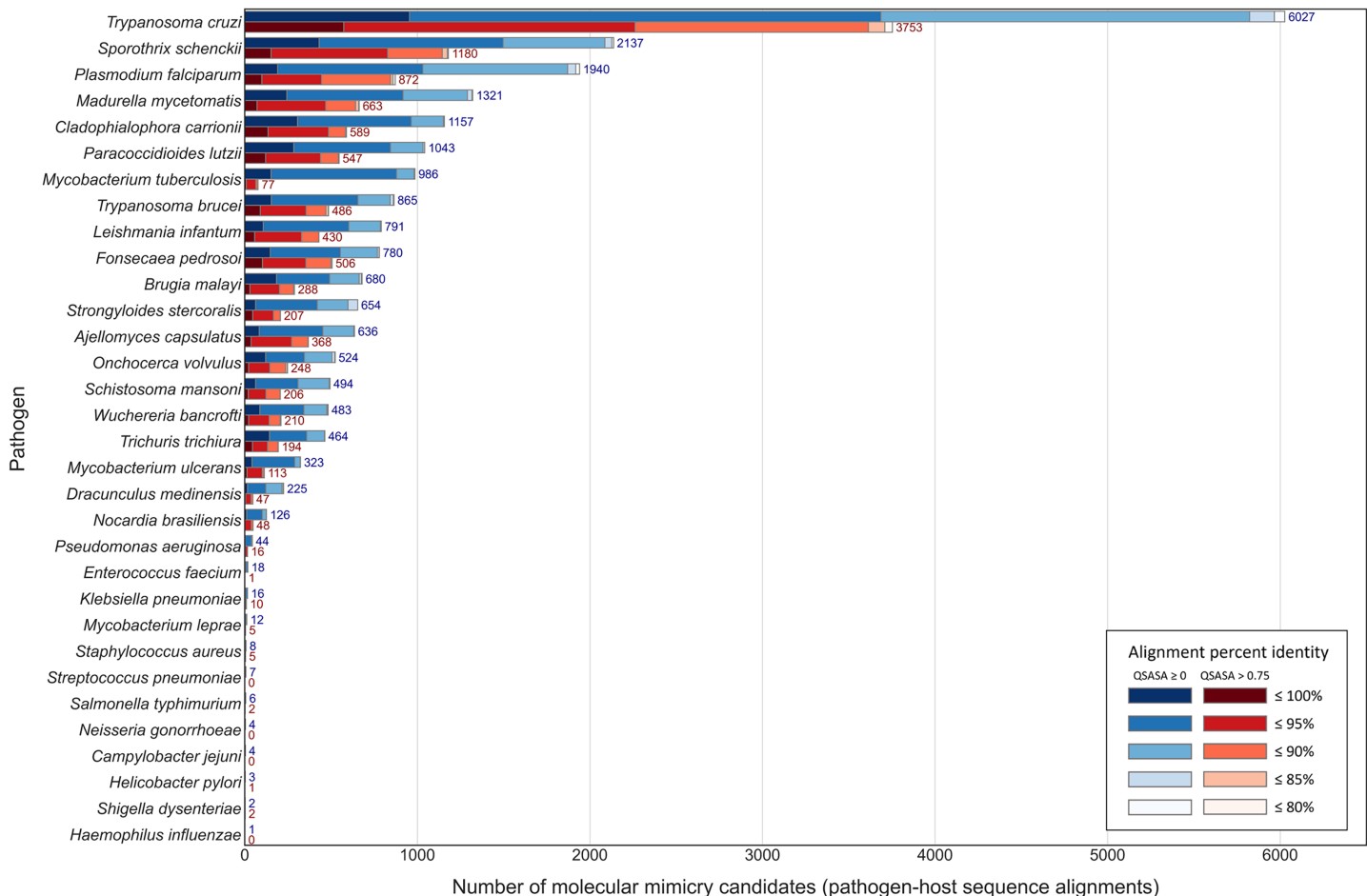

**Figure 5** **Number of paired pathogen-host regions before (blue) and after (red) QSASA filtering, coloured by alignment percent identity.**

1,439 proteins, we found that 51% of the clusters were paired to a single pathogen species, 18% were paired to two pathogen species, and 17% were paired to five or more pathogen species (Fig. 7). Similarly, when the human proteins were clustered at 40% identity, resulting in 1,254 clusters for the 1,439 proteins, we found that 48% of the clusters were paired to a single pathogen species, 17% were paired to two pathogen species, and 20% were paired to five or more pathogen species. This suggests that while multiple members of a given protein family may be targeted by different pathogens, the total number of protein families targeted is large and diverse at the sequence level.

We conducted an analysis to identify recurring biological processes by looking for enrichment of Gene Ontology (GO) terms in the proteins containing MHRs. We identified a total of 418 enriched GO terms, with 413 of them being enriched in three or more pathogen species (Table S1). Among these 418 terms, 343 were positively associated, while 75 were negatively associated with MHRs. Within these enriched GO terms, we focused on those that we thought would likely be involved in host-pathogen interactions (Table 2).

We found that the term 'immune system process' (GO:0002376) was associated with 126 human proteins containing MHRs, with proteins from 22 different pathogen species.

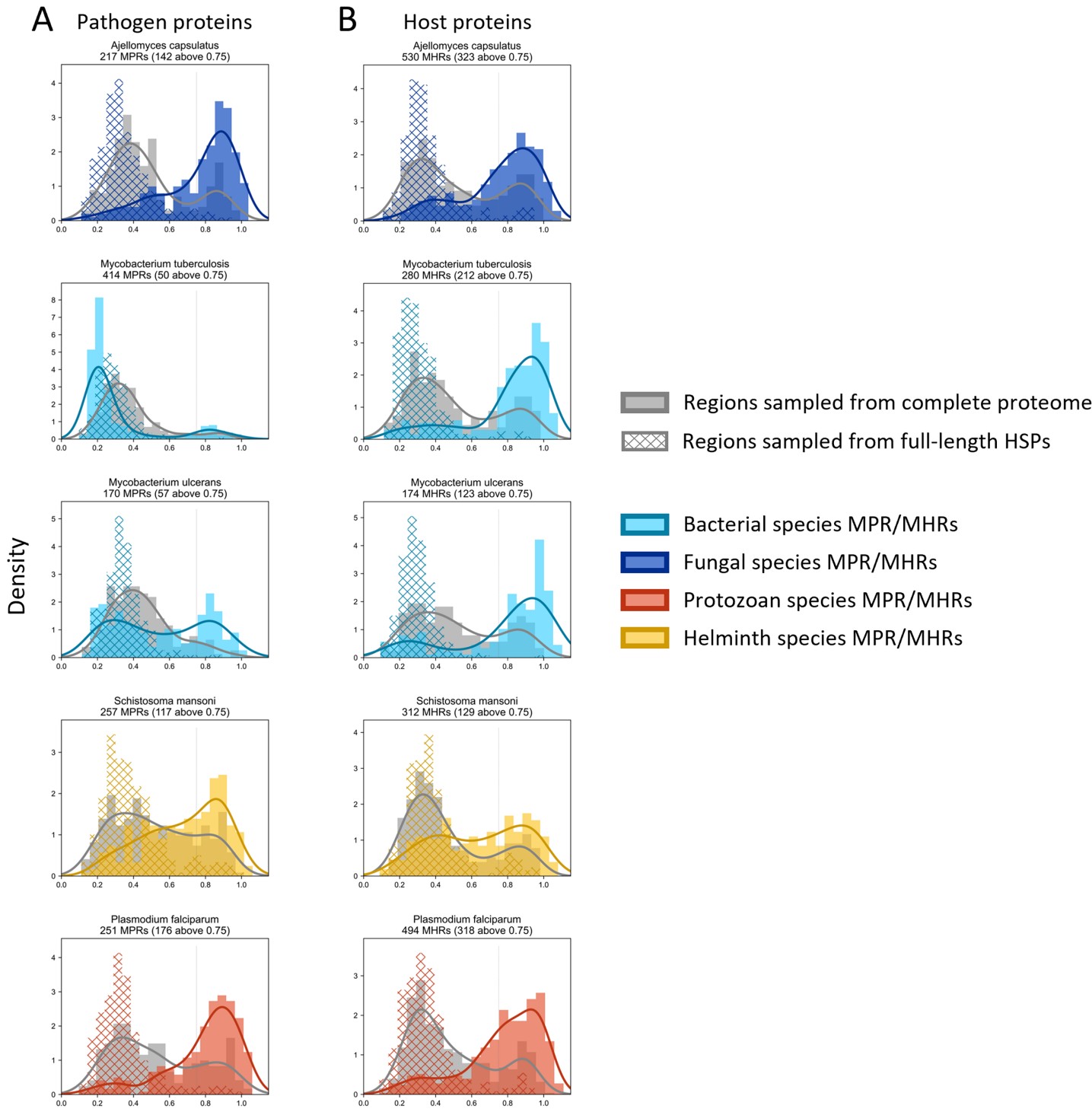

**Figure 6** **Solvent accessibility of potential mimicry regions for selected pathogens.** Histograms/KDE plots for average solvent accessibility of potential mimicry candidates from select pathogen species (A) and human proteins (B) with equivalent-length regions randomly selected from the complete pathogen proteome (grey) and from full-length protein HSPs (hatched). Vertical grey lines indicate 0.75 QSASA threshold. The plots for all pathogen-host pairs are shown in Figs. S1 and S2, and at https://github.com/Kayleerich/molecularmimicry/.

**Table 2 Discussed and related gene ontology terms ($p < 0.001$, FDR < 0.05).**

| GO term ID | Description | FE | $p$-value | FDR | Human proteins | Pathogen species |
|---|---|---|---|---|---|---|
| GO:0002250 | Adaptive immune response | 0.25 | 8.38E−09 | 1.08E−06 | 11 | 15 |
| GO:0002376 | Immune system process | 0.74 | 4.49E−04 | 2.13E−02 | 126 | 22 |
| GO:0002443 | Leukocyte mediated immunity | 0.27 | 1.43E−04 | 7.97E−03 | 6 | 7 |
| GO:0002449 | Lymphocyte mediated immunity | 0.28 | 6.42E−04 | 2.83E−02 | 5 | 7 |
| GO:0006955 | Immune response | 0.48 | 7.19E−10 | 1.09E−07 | 53 | 18 |
| GO:0006959 | Humoral immune response | 0.38 | 1.36E−03 | 4.95E−02 | 9 | 11 |
| GO:0007010 | Cytoskeleton organization | 1.42 | 2.12E−04 | 0.011 | 126 | 24 |
| GO:0016043 | Cellular component organization | 1.30348 | 2.27E−11 | 3.70E−09 | 517 | 25 |
| GO:0019724 | B cell mediated immunity | 0.22 | 1.35E−03 | 4.95E−02 | 3 | 3 |
| GO:0030198 | Extracellular matrix organization | 2.32 | 9.73E−07 | 9.80E−06 | 47 | 20 |
| GO:0030595 | Leukocyte chemotaxis | 0.1 | 7.37E−04 | 3.13E−02 | 1 | 9 |
| GO:0030851 | Granulocyte differentiation | 5.4 | 1.94E−04 | 1.05E−02 | 9 | 9 |
| GO:0034063 | Stress granule assembly | 4.97 | 3.21E−04 | 0.016 | 9 | 10 |
| GO:0038065 | Collagen-activated signaling pathway | 5.68 | 7.91E−04 | 0.033 | 7 | 8 |
| GO:0043207 | Response to external biotic stimulus | 0.6374 | 1.40E−04 | 7.83E−03 | 64 | 18 |
| GO:0045087 | Innate immune response | 0.55 | 3.14E−04 | 1.57E−02 | 32 | 18 |
| GO:0050851 | Antigen receptor-mediated signaling pathway | 0.25 | 1.14E−03 | 4.40E−02 | 4 | 4 |
| GO:0098542 | Defense response to other organism | 0.5362 | 1.93E−05 | 1.42E−03 | 40 | 18 |

However, all these terms were negatively associated with the MHRs, indicated by fold enrichment values less than 1. Other enriched terms in our MHR set, such as 'defense response to other organism' (GO:0098542) and 'response to external biotic stimulus' (GO:0043207), were also negatively associated with the MHRs. The only enriched GO term related to immune function that showed a positive association with MHRs was 'granulocyte differentiation' (GO:0030851), which is a child term of 'leukocyte differentiation' (GO:0002521). This term was assigned to MHRs in proteins from nine pathogen species, including four fungi, one bacterium, and two protozoans, of which seven are intracellular pathogens. We also looked at enriched GO terms related to 'cellular component organization' (GO:0016043). We found 47 human proteins annotated 'extracellular matrix organization' (GO:0030198) were mimicked by 20 pathogen species, including proteins Amyloid-beta precursor protein (P05067) and Elastin (P15502). Each shared MHRs with proteins from four pathogen species: P05067 with two fungi, two protozoans; and P15502 with two helminths, two protozoans. The human collagens were targeted by 23 pathogen species. A subset, annotated as 'collagen-activated signaling pathway' (GO:0038065), included collagens that are cleaved to create Arresten (P02462) and Canstatin (P08572), which inhibit endothelial cell proliferation and migration (*Kamphaus et al., 2000*; *Nyberg et al., 2008*), shared MHRs with three helminth and one fungus species. Also within this subset was human platelet glycoprotein VI (Q9HCN6) which plays a role in procoagulation and wound-healing (*Jandrot-Perrus et al., 2000*).

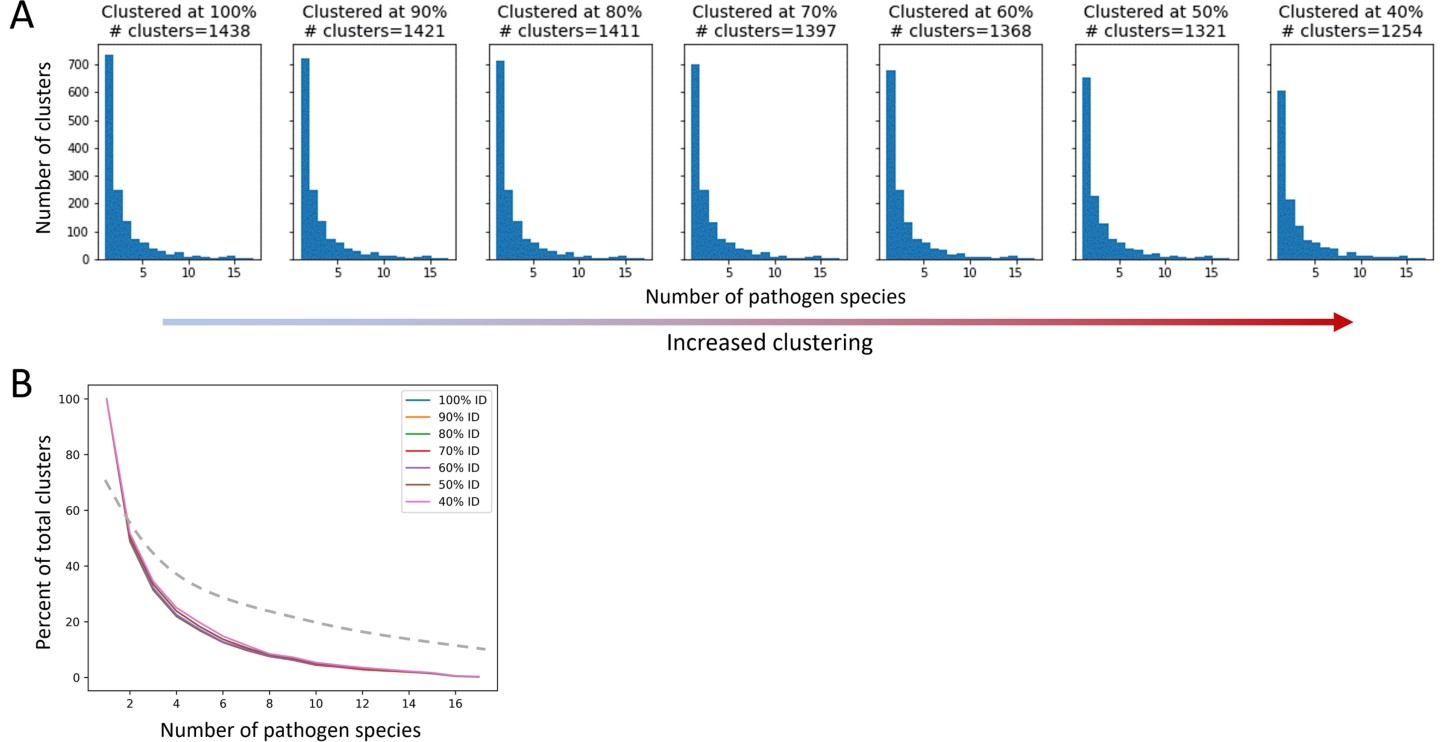

**Figure 7** **The proportion of host protein clusters associated with more than one pathogen species does not change with increased clustering.** A cluster is considered associated with a pathogen species if a protein sequence from the species paired to at least one human sequence in the cluster. (A) Total number of host protein clusters associated with one or more pathogen species at each percent identity clustering value. Clustering increases from left to right. (B) Proportion of total clusters associated with one or more pathogen species for each clustering value. The dashed line represents an example of expected proportions for increased clustering if multiple pathogens were mimicking similar host proteins.

We found that 24 pathogen species targeted 126 human proteins that were annotated with the term 'cytoskeleton organization' (GO:0007010). These human proteins included human Type I & II Keratins, which are involved in epithelial cell intermediate filament formation (*Jacob et al., 2018*), Ankyrin-3 domain containing proteins (Q12955), which regulate cytoskeleton anchoring (*Bennett & Baines, 2001*), and Ataxin proteins (Q99700 and Q8WWM7) which mediate actin stabilisation (*Del Castillo et al., 2022*).

We investigated the extent to which LCRs featured in our hunt for molecular mimics and whether they confounded our results (Fig. 8). Prior to LCR filtering, we found that a single MHR may align to more than a hundred pathogen proteins from multiple pathogen species. For example, the human protein KAT6B (Q8WYB5) contains two MHRs, one of which aligned to 120 protein regions from fifteen pathogen species. Out of a total 1,438 human proteins, we identified 1,145 that contained a single MHR, 212 that contained two MHRs, and 81 that contained three or more MHRs–up to a maximum of ten MHRs from a single protein (Fig. 9). When we removed any potential mimic that overlapped an LCR, the total number of MHRs decreased to from 1,878 to 76. For 14 pathogen species (all bacteria), this step removed every MHR. For other species the number of results was greatly reduced. For example, in *P. falciparum*, we originally found 229 MHRs which fell to one when we removed any MHR that contained an LCR. To us, this filter was too extreme.

```
            Human (Q9BVH7)   33 ERPPQQQQQQQQQQQQASAT 52
   Leishmania infantum (A4HTL3)  249 EAPPQQQQQQQQE·······  262
   Leishmania infantum (A4HSU7)  123 ······QQQQQQQQQQQASAY 138
Madurella mycetomatis (A0A175W1X9)  89 ·····QQQQQQQQQQQQLSA· 104
   Sporothrix schenckii (U7PUD6)  393 ······EQQQQQQQQQQAAAT 407
       Trypanosoma cruzi (Q4CW81)   21 ······QQQQQQQQQLTSAT 35
       Trypanosoma cruzi (Q4DYA1)   21 ······QQQQQQQQQLTSAT 35
```

**Figure 8 Example of low complexity in mimicry regions.** The MHR from Q9BVH7 overlaps an LCR by 60% and the LCR overlap is not necessarily equal length between pathogen and human proteins. Proteins are referred to by Uniprot ID, mismatched residues from MPR to MHR are coloured grey.

Therefore, we only removed MHRs for which more than 50% of their length was low complexity. By this filtering, we found 100 MHRs in 98 proteins. Two proteins contained two MHRs, with each MHR only associated with one species. Of the total 100, no MHR was associated with more than two pathogen species, and only six MHRs were associated with two species. We also found that 86 of the 100 LCR-filtered MHRs overlapped a disordered region of the protein, and that 78% of the MHR residues were predicted to be disordered (IUPRED score > 0.50), as compared to 22% of all residues in the human proteome.

We analyzed this reduced set of human proteins again using Gene Ontology terms and found only three enriched terms: "extracellular matrix organization" (GO:0030198) and two ancestor terms "extracellular structure organization" (GO:0043062) and "external encapsulating structure organization" (GO:0045229). Pathogens from all four major taxonomic groups had proteins that mimicked human proteins with these terms. Of the proteins that were annotated with all three of the above GO terms, we decided to explore two in more detail.

We found four *P. falciparum* proteins that paired to the same region of the human protein vitronectin (VTNC_HUMAN, Uniprot: P04004) (Fig. 10A). When we queried full-length *Plasmodium* proteins against the human proteome (BLASTP, default parameters), vitronectin was not a significant result (E-value > 0.001). When we aligned the *P. falciparum* full-length proteins to vitronectin using MAFFT, the MPRs and MHR did not align when the progressive method FFT-NS-1 was used but did align using the iterative method L-INS-I. When we aligned them to vitronectin individually, the MPRs of each aligned to the vitronectin MHR for three of the four parasite proteins. This indicates that this mimicry region may be identified when individual proteins are aligned, but these mimicry candidates would not be identified by querying full-length parasite proteins using BLASTP.

Vitronectin interacts with the VTNC receptor (αvβ3 integrin) expressed on platelets *via* an RGD motif at 64-66. It is found in human plasma, where it is involved in cell adhesion (*Schvartz, Seger & Shaltiel, 1999*) and can be protective against complement lysis (*Milis et al., 1993*). All four of the *P. falciparum* proteins are part of the *Plasmodium falciparum* erythrocyte membrane protein 1 (PfEMP1) family and contain a CIDR1-γ domain which has been associated with rosetting of erythrocytes during severe malaria infections (*Vigan-Womas et al., 2012*). Some proteins from this family are inserted into the membrane of

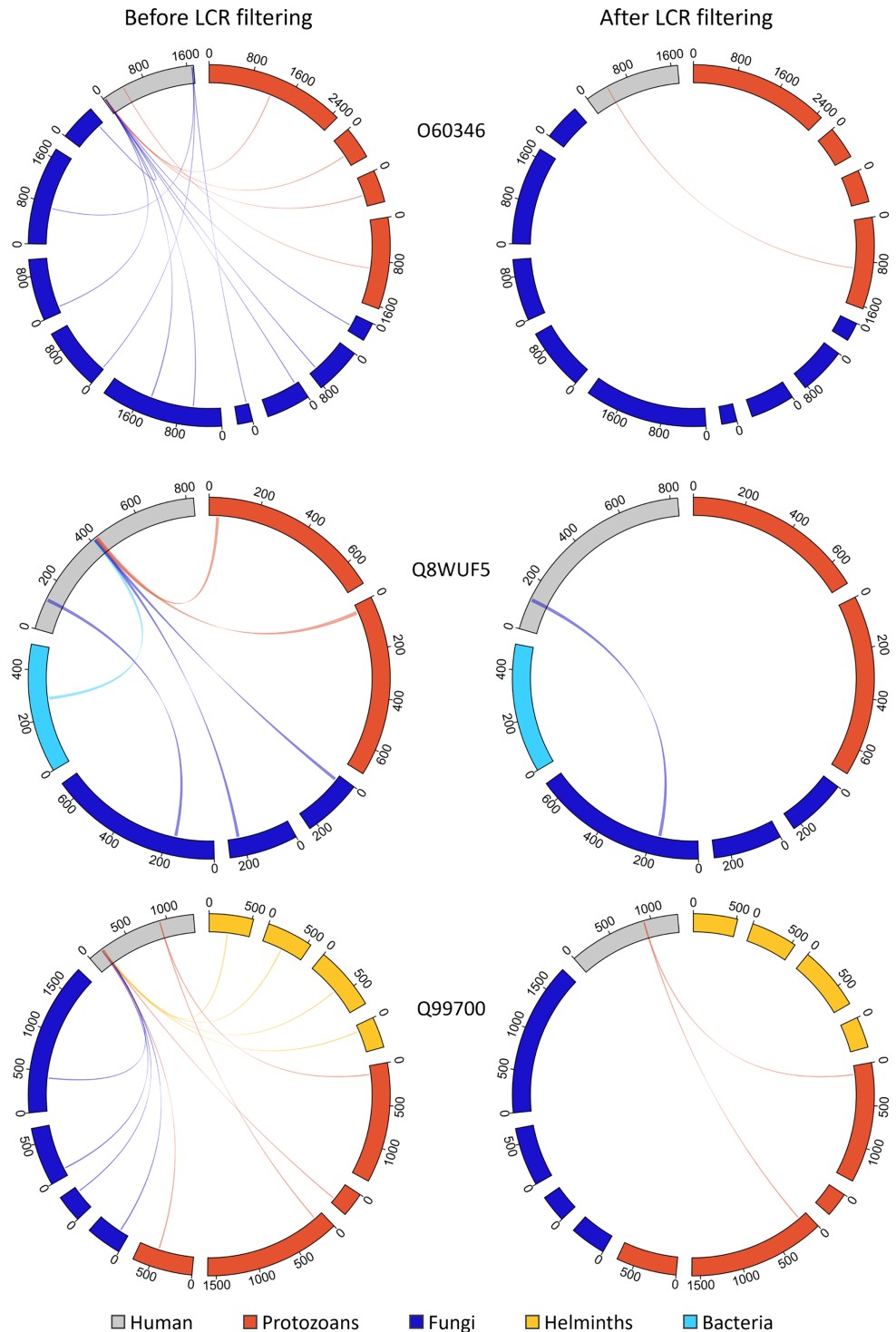

**Figure 9 Removing MHRs comprised of more than 50% LCR removes most MHRs which were aligned to MPRs from more than one pathogen species.** In these examples, the human proteins contain multiple MHRs. O60346 contains three MHRs which pair with up to ten MPRs from five pathogen species, Q8WUF5 contains two MHRs paired with up to five MPRs from five pathogen species, and Q99700 contains three MHRs paired with up to eight MPRs from seven pathogen species. After filtering out MHRs comprised of 50% or more LCR, the number of MHRs drops to one for all human proteins and eliminates all MHRs paired to more than two MPRs.

**Figure 10 Alignments of select mimicry candidates.** (A) Alignment of identified MPRs from four *P. falciparum* PfEMP1 proteins to the MHR of vitronectin. (B) Alignment of identified MPRs from two *T. cruzi* proteins to human ADAM15. Dashed boxes indicate the individual pathogen regions identified.

infected erythrocytes, where intracellular portions can interact with the host cytoskeleton and extracellular portions are involved in cell adhesion. The PfEMP1 MPRs identified are on extracellular portions of the proteins, and two of the PfEMP1 proteins contain an extracellular RGD motif. The MHR of vitronectin has no annotation and is predicted as disordered.

We identified MPRs from two uncharacterized *T. cruzi* proteins (Uniprot: Q4DU61_TRYCC and Q4DLR3_TRYCC) which shared an MHR on the protein ADAM15 (ADA15_HUMAN, Uniprot: Q13444) (Fig. 10B). When we aligned the full-length proteins, the identified MPRs did not align to the MHR on ADAM15. We also queried the full-length sequences of ADAM15 and the two *T. cruzi* proteins against the full proteomes of *T. cruzi* and Human, respectively, and found no significant results (BLASTP default settings, e-value < 0.001).

The ADAM15 MHR is found on its cytoplasmic tail between two SH3-interaction motifs. ADAM15 has been associated with wound healing (*Charrier et al., 2005*) and cell migration (*Herren et al., 2001*). There are 13 reported isoforms of ADAM15 on Uniprot that have varying isoform-specific interactions between the cytoplasmic tail and SH3 domain-containing proteins (*Zhong et al., 2008*; *Mattern et al., 2019*). The MHR we identified is found on the most common, ubiquitously expressed isoform. Other isoforms containing the MHR are expressed in various tissues and are highly expressed in peripheral leukocytes (*Kleino, Ortiz & Huovila, 2007*). We further analyzed the two *T. cruzi* proteins using LMDIPred, a web server for prediction of probable SH3, WW, and PDZ binding sites (*Sarkar, Jana & Saha, 2018*), and found predicted SH3-binding motifs within 50 residues of the identified MPRs.

## DISCUSSION

Identification of novel molecular mimicry candidates improves our understanding of pathogen-host interactions with the potential to inform the development of future therapeutic strategies. Here, we code for a previously described algorithm, which we extended to include additional filtering, and search for molecular mimicry candidates across a selection of pathogenic prokaryotic and eukaryotic species' proteomes. We found
the most mimicry candidates in pathogenic fungi and protozoans. Additionally, we were surprised by two observations: (i) helminths and bacteria had relatively few mimicry candidates; (ii) proteins annotated as involved in immune processes were significantly under-represented in mimicry candidates in all taxonomic groups.

Helminths are known to modulate their hosts' immune systems through multiple types of biomolecules (reviewed in *Zakeri et al., 2018*). In one example, *Heligmosomoides bakeri* (at the time misclassified as *Heligmosomoides polygyrus* (*Cable et al., 2006*; *Stevens et al., 2023*)) secretes a protein which interferes with the TGF-β receptors TβRI and TβRII by binding at sites distinct from the sites used by TGF-β. This dampens the host's immune response against the parasite. The protein was named TGF-b mimic (TGM) (*Johnston et al., 2017*), but Hb-TGM shares neither sequence nor structural similarity with TGF-β. Although Hb-TGM could be considered a functional mimic, the absence of shared epitopes excludes it from accepted definitions of molecular mimicry. The molecular interactions between many helminth species and their hosts are also mediated by parasite-encoded small non-coding microRNAs (miRNA), which are packed into extracellular vesicles and released into the host (reviewed in *Rojas-Pirela et al., 2022*). Rapidly after infection, *Fasciola hepatica* secretes a miRNA which mimics the host's miR-125b and interrupts MAPK signalling so reducing the innate immune responses (*Tran et al., 2021*). A homologue to *F. hepatica* miR-125b could be detected in *S. mansoni*, a species included in our study. Sequencing the RNA secreted from *H. bakeri* identified at least 18 miRNA that shared significant similarity with miRNA from its mouse host (*Hambrook & Hanington, 2021*). While these examples confirm that mimicry is a feature of helminth-host interactions, mimicry is not limited to proteins. Non-proteinaceous mimicry can involve RNAs (*Tran et al., 2021*), lipids (*Laan et al., 2017*), secondary metabolites (*Ekanayake, Skog & Asp, 2007*; *Rubenstein, 2008*), and carbohydrates (*Ang, Jacobs & Laman, 2004*; *Hirayama et al., 2007*; *van Die & Cummings, 2010*; *Kappler & Hennet, 2020*) which are excluded from our protein-based searches. Similarly, most known modulators of bacteria-host interactions would not be detected in our pipeline. This does not preclude molecular mimicry from being an important evolutionary mechanism. Further, there may be technical reasons, which are discussed further below.

An early prediction of ours was that molecular mimicry offered a promising evolutionary mechanism by which to disrupt immune function. This assumption was incorrect. Mimicry requires sequence or structural similarity with a host protein, which facilitates a specific pathogen-host interaction. Specific interactions are not ideal for general immune interference, and pathogens favour more versatile tactics such as avoidance or disruptive strategies. For instance, *Staphylococcus aureus* avoids host defensins, which are attracted to negatively-charged bacterial membrane lipids. The bacterial multiple peptide resistance factor protein (MprF) modifies the outer membrane lipids to neutralise the negative charge, which repels defensins (*Ernst et al., 2009*). As a general disruption strategy, *Yersinia* spp. secrete an acetyltransferase, *Yersinia* outer protein J (YopJ), which modifies a variety of residues on the activation loop of host MAPKKs and IKKβ, blocking phosphorylation by upstream kinases and inhibits signaling to downstream immune pathways (*Mukherjee, Hao & Orth, 2007*; *Paquette et al., 2012*).

YopJ shares no significant sequence similarity with a metazoan protein, so is not an example of molecular mimicry.

We did find an overrepresentation of mimicked proteins involved in cytoskeleton and cellular adhesion. Cytoskeletal proteins are common targets for molecular mimicry. One example is *Listeria monocytogenes* which facilitates interaction with the host cytoskeleton and motility of the bacterium within the host cell by mimicking Wiskott–Aldrich syndrome (WAS) family proteins. The N-WASP-mimicking *L. monocytogenes* protein, actin assembly-inducing protein (ActA), contains a region of 31 amino acids that interacts with the human actin-related protein complex (Arp2/3), hijacking the actin nucleation process to propel itself through the host cell (*Zalevsky, Grigorova & Mullins, 2001*; *Zalevsky et al., 2001*). Cell adhesion and interactions with the extracellular matrix may also require similarity to a host protein for specificity of function. As previously mentioned, *C. albicans* and *H. pylori* exploit extracellular adhesion molecules to further their infection and invasion of host cells. With these, and other, examples in mind, we investigated the biological processes associated with our results. We found potential mimics of cytoskeletal processes across all groups in our study. Similarly, we found potential mimics targeting extracellular matrix organization across all groups, even after removing LCR content.

Many of the mimicking regions we found contained low complexity regions (LCRs). These LCRs diverge from the expected amino acid composition—perhaps towards a specific amino acid (*e.g.*, homorepeats) or a type of amino acids (*e.g.*, highly acidic). They are highly prevalent in eukaryote proteomes and many are involved with the interaction between their protein and other biomolecules, *e.g.*, RNA, DNA, and other proteins (*Crane-Robinson, Dragan & Privalov, 2006*). It is, therefore, no surprise that LCRs mediate host-pathogen interactions (*Mier & Andrade-Navarro, 2021*). In the malaria-causing *Plasmodium vivax*, LCRs within a surface protein may assist with impairing antigen-antibody binding (*Kebede et al., 2019*). LCRs are much less prevalent in bacteria, but are highly conserved in extracellular and outer membrane proteins in pathogenic strains (*Mier & Andrade-Navarro, 2021*). Considering this conservation and that mutation rates increase with proximity to LCRs (*Huntley & Clark, 2007*; *Haerty & Golding, 2010*; *McDonald et al., 2011*; *Jovelin & Cutter, 2013*; *Lenz, Haerty & Golding, 2014*), it is likely that host regulatory LCRs are mimicked by pathogens. However, it is incredibly challenging to identify the function of an LCR. Given their repetitive nature and widespread use across a broad taxonomic range, we anticipated that our comparison of pathogen 14-mers to the negative control set would eliminate virtually all LCRs from consideration. The presence of LCRs in our final set raises questions about their biological relevance in the context of molecular mimicry and our technical approach.

While we were investigating the LCR content of the MHRs, we observed that a large proportion of the LCR-filtered MHRs were in intrinsically disordered regions of the protein. A protein may be considered disordered if it does not form a stable structure at physiological conditions (*Dyson & Wright, 2005*; *Habchi et al., 2014*). The structural

flexibility of intrinsically disordered regions allows them to interact with a large array of protein partners (*Tompa, Szász & Buday, 2005*), but also makes prediction of their tertiary structures particularly difficult (*Ruff & Pappu, 2021*). The search for molecular mimicry is increasingly looking at structural similarities, and while this is a natural extension for the pipeline present here, our work and that of others demonstrates that careful curation of the datasets and results is crucial (*Guven-Maiorov et al., 2020*; *Muthye & Wasmuth, 2023*; *Balbin et al., 2023*). Intrinsically disordered regions are prone to low-confidence scores during structural prediction and are, therefore, predisposed to be missed by mimicry methods that rely on structural similarity. This underlines the importance of including primary sequence comparisons for molecular mimicry identification.

Finally, we want to offer a piece of advice for others wishing to use ours or similar approaches. The choice of negative control species will have a large effect on mimicry detection. In our study, we were limited by the criterion that three-dimensional structures should be available for the full proteomes of all species. At the time of study, AlphaFold2 predictions were available for model organisms. We excluded *Mus musculus* and *Rattus norvegicus* from the control set as a pathogen's mimicry to a human protein is likely to be shared with other mammals. As the structural predictions for most species in UniProt become available, the choice of control species can be refined. We recommend this set is tailored and include non-pathogenic species or strains closely related to the pathogen under scrutiny. For example, inclusion of the non-pathogenic *Mycobacterium indicus pranii* and opportunistic *Mycobacterium intracellulare* would assist with understanding virulence in tuberculosis (*Rahman et al., 2014*).

## CONCLUSIONS

Overall, we identified molecular mimicry candidates between proteins from 28 pathogen species and human proteins. From these candidates, we identified commonalities in human signalling pathways targeted through mimicry. We also included the addition of QSASA filtering and LCR removal to narrow our search beyond sequence similarity, but this reduced the number of results drastically. It is unlikely that true mimics were removed with solvent-inaccessible regions, so there is a concern that true mimics were rejected during sequence filtering. In this regard, we have identified areas of concern with potential for improvement: consideration of the biochemical properties for mismatched residues and parameter optimization. Nevertheless, we assert that sequence alignments are valuable to the pursuit of accurate computational mimicry identification and provide accessible tools to aid in this endeavour.

## ACKNOWLEDGEMENTS

We acknowledge the high-performance computing resources made available by the Faculty of Veterinary Medicine and Research Computing at the University of Calgary. We thank the anonymous reviewers for their helpful comments.

### Funding

This work is supported by the Natural Sciences and Engineering Research Council of Canada (NSERC) Discovery Grants (No. 04589-2020) to James D. Wasmuth and University of Calgary Eyes High scholarships to Kaylee D. Rich and Viraj R. Muthye. The funders had no role in study design, data collection and analysis, decision to publish, or preparation of the manuscript.

### Grant Disclosures

The following grant information was disclosed by the authors:
Natural Sciences and Engineering Research Council of Canada (NSERC): 04589-2020.
University of Calgary Eyes High doctoral student recruitment scholarship.
University of Calgary Eyes High postdoctoral recruitment scholarship.

### Competing Interests

The authors declare that they have no competing interests.

### Author Contributions

- Kaylee D. Rich conceived and designed the experiments, performed the experiments, analyzed the data, prepared figures and/or tables, authored or reviewed drafts of the article, and approved the final draft.
- Shruti Srivastava conceived and designed the experiments, performed the experiments, authored or reviewed drafts of the article, and approved the final draft.
- Viraj R. Muthye conceived and designed the experiments, authored or reviewed drafts of the article, and approved the final draft.
- James D. Wasmuth conceived and designed the experiments, analyzed the data, prepared figures and/or tables, authored or reviewed drafts of the article, and approved the final draft.

### Data Availability

All Python and Bash scripts used are available at GitHub and Zenodo:
- https://github.com/Kayleerich/molecularmimicry.
- Kaylee Rich (2023). Kayleerich/molecularmimicry: MMv1.0 (molecularmimicry). Zenodo. https://doi.org/10.5281/zenodo.8361283.

### Supplemental Information

Supplemental information for this article can be found online at http://dx.doi.org/10.7717/peerj.16339#supplemental-information.

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
