# Peer review of "Identification of potential molecular mimicry in pathogen-host interactions"

_PeerJ, doi:10.7717/peerj.16339_

## Round 0.1 · original submission · Major Revisions

The two reviewers were quite mixed in their response to your work. Please address reviewer 1's comments and also address this comment from reviewer 2: "This manuscript is just a recreation of sequence-based bioinformatics pipeline as described by Ludin et al. (2011). But there is no benchmark results to show how their results compared to the original one."

Reviewer 1 ·

Basic reporting

English: I found the writing to be generally clear. There were several typographical errors, which are annotated in the attached PDF. Please see editorial notes not only on the manuscript body, but also in the References section and in the figure and table legends.

The Discussion section could be improved by a transition sentence at Line 290, that clarifies that the authors have two general explanations/examples for why some pathogens had few mimicry candidates and why immune processes were under-represented. In this part of the discussion, the authors might help readers by clearly disambiguating mimicry from avoidance (e.g., of immune recognition) and disruption (e.g., of signaling processes or of immune responses or of defense responses). Only the first (molecular mimicry) would be expected to require protein homology with the host. The authors might also reiterate, in the first paragraph of the discussion, other types of molecules that could be involved in mimicry (as well as avoidance or disruption) that are NOT proteins: lipids, secondary metabolites, RNAs, and carbohydrates. Mention of extracellular vesicle delivery of these entities would be useful in the first paragraph. That could set the reader up to better follow the arguments and examples below.

At line 311-313, the sentence could be better written to explain that immune avoidance is essentially different from mimicry, in that it involves changing or hiding exposed bacterial epitopes so that they are unrecognized by immune cells. Two examples could be phase variation and capsules.
In the paragraph at lines 351-363, I was looking for some wording to explain that LCRs, while perhaps “commonly used tools”, and therefore commonly encountered, are likely context-dependent in their utility. But perhaps I don’t understand this concept well enough to make a suggestion here!
Literature: References seem sufficient and current.

Figures, tables, data: Figures and tables were well constructed, easy to read and interpret, and useful in visualizing data. I found the source code for the pipeline at the URL listed in the methods section. Uniprot IDs are listed in Table 1, for the 32 pathogens, 13 control species, and 1 host. Alignment results were not made available.

Self-contained: This work represents a complete story. It offers a new pipeline tool for analyzing genomes for potential mimicry, and ground-truths the tool on a large pathogen dataset. It then describes strengths and limitations of the tool in understanding molecular host-pathogen interactions.

Experimental design

Aims and Scope: This article falls into the category of Bioinformatics Software Tools. It is scientifically sound.

The research goals are clearly defined: to create a Python pipeline to search for similarity between human and pathogen proteins; and to search for proteins that are found in pathogens and which potentially mimic host proteins among a large number (32) of human pathogens.

Rigorous investigation: Here I will list my only substantial concern with this paper. It is not clear to me how the pathogens and controls were chosen. Some bias may have been introduced by including three species of Mycobacterium among the bacterial pathogens; Trypanosoma is also represented twice. It is not clear to me why four plants, zebrafish, the fruit fly, and a thermophilic methanogen were included among the controls. These are indeed phylogenetically distant from the pathogens and from the host… but I was left wondering why the authors didn’t (instead) choose an approach where controls were phylogenetically closer to the pathogens, but representing non-pathogenic lifestyles. Another way to say this is that the controls should represent a “core genome” common to a clade of organisms that include the pathogens, and the pathogen genes should differ from the core genome. I think that the study is valuable the way it was done, but would appreciate a more thorough explanation of the logical process used to decide upon which genomes to include. This logical process could either be described in the Introduction or in the Methods.

Methods: I believe that the Methods were described in sufficient detail for someone knowledgeable in bioinformatics to replicate.

Validity of the findings

My only concern is the one stated in the section above regarding choice of genomes as “pathogens” and “controls”; since this is fundamental to the interpretation of the results from testing the pipeline, I feel that the authors should provide more rationale as to their choice of subject genomes.

Additional comments

I really enjoyed reading this paper and am eager to see it published. I believe the tools that the authors have developed will be very useful to work on host-microbe interactions. Although viruses were not included in the test data, the utility of this tool for virology is obvious as well.

Annotated reviews are not available for download in order to protect the identity of reviewers who chose to remain anonymous.

Reviewer 2 ·

Basic reporting

Generally, the image quality is low. Please increase the resolution. Also, too many figures (11 figures) in the main texts.

Moreover, I think the writing is not structured well.

For example,

No informative and quantitative results were reported in the abstract. In line 21-25: is the results statistically significant? How many pathways were retrieved to be involved with the host extracellular matrix and cytoskeletal processes?

The introduction sections are written in a logic flow that (1) what is molecular mimicry and 2 common consequences, (2) immune modulation is also responsible for other host-pathogen interactions, (3) the current gap that conventional way to identify molecular mimicry is
Suggestions to Introduction
1. Line 39-42: only 1 reference is cited but no reference for "These two consequences
42 should be considered to exist on a spectrum, rather than independently."

Experimental design

This manuscript is just a recreation of sequence-based bioinformatics pipeline as described by Ludin et al. (2011). But there is no benchmark results to show how their results compared to the original one.

Validity of the findings

The novelty is not assessed. And the conclusions are not well stated in a meaningful and quantitative manner.

---

## Round 0.2 · Minor Revisions

Your revision is much improved but I suggest you follow up on the minor suggestions from reviewer 1. I will not have to send this back to the reviewers.

Reviewer 1 ·

Basic reporting

See previous review

Experimental design

See previous review

Validity of the findings

See previous review

Additional comments

This is my second review of the manuscript. I find that the authors have addressed my initial questions in a satisfactory manner. I now have minor comments:

Please use and describe the acronym LCR the first time low complexity regions are described, in the final sentence of the Introduction.

In the Discussion, the placement of the argument in Lines 329-333 (Word Track Changes doc) is awkward. It might make better logical sense placed after line 359, as an explanation for why the helminths in this study had few host-mimicking proteins.

In the penultimate line of the discussion, the authors question whether BLASTP is appropriate for this method, which is a surprise, undermining our trust in the data. They offer no further explanation. I suggest expanding upon this.

There are several tantalizing results whose biological relevance could be expanded in the Discussion if the authors choose:
-importance of lysosome-dependent pathogen entry upon the abundance of solvent-accessible host mimicking regions (e.g., are pathogen proteins intact in the pathogen-containing vacuoles?)
-medical relevance of amyloid-beta precursor protein discovery as a mimic in four pathogens
-in M. tuberculosis, does the lack of solvent-accessible regions acting as host mimics perhaps correlate with its waxy, hydrophobic envelope?

Finally, I suggest being up-front in the abstract about the nature of the study. It might be useful to state directly, in the second sentence of the abstract: "In this study, we resurrected, updated, and optimized a sequence-based bioinformatics pipeline to identify potential molecular mimicry candidates between humans and 32 pathogenic species whose proteomes' 3D structure predictions were available at the time of the study's inception."

Reviewer 2 ·

Basic reporting

The revised version has improved the clarity and has successfully conveyed its novelty compared to last version.

Experimental design

Overall, the structure of the flow chart is well designed

Validity of the findings

All underlying data have been provided.

---

## Round 0.3 · accepted · Accept

You have addressed the reviewer's comments and your manuscript is ready for publication, congrats!